# Chloride Channels and Transporters: Roles beyond Classical Cellular Homeostatic pH or Ion Balance in Cancers

**DOI:** 10.3390/cancers14040856

**Published:** 2022-02-09

**Authors:** Hyeong Jae Kim, Peter Chang-Whan Lee, Jeong Hee Hong

**Affiliations:** 1Department of Physiology, Lee Gil Ya Cancer and Diabetes Institute, Gachon University, 155 Getbeolro, Yeonsu-gu, Incheon 21999, Korea; lilili1125@naver.com; 2Department of Biomedical Sciences, Lung Cancer Research Center, University of Ulsan College of Medicine, Asan Medical Center, Seoul 05505, Korea

**Keywords:** chloride channels, chloride-associated transporters, prognostic marker, metastasis, migration and invasion

## Abstract

**Simple Summary:**

Roles of chloride-associated transporters have been raised in various cancers. Although complicated ion movements, crosstalk among channels/transporters through homeostatic electric regulation, difficulties with experimental implementation such as activity measurement of intracellular location were disturbed to verify the precise modulation of channels/transporters, recently defined cancerous function and communication with tumor microenvironment of chloride channels/transporters should be highlighted beyond classical homeostatic ion balance. Chloride-associated transporters as membrane-associated components of chloride movement, regulations of transmembrane member 16A, calcium-activated chloride channel regulators, transmembrane member 206, chloride intracellular channels, voltage-gated chloride channels, cystic fibrosis transmembrane conductance regulator, voltage-dependent anion channel, volume-regulated anion channel, and chloride-bicarbonate exchangers are discussed.

**Abstract:**

The canonical roles of chloride channels and chloride-associated transporters have been physiologically determined; these roles include the maintenance of membrane potential, pH balance, and volume regulation and subsequent cellular functions such as autophagy and cellular proliferative processes. However, chloride channels/transporters also play other roles, beyond these classical function, in cancerous tissues and under specific conditions. Here, we focused on the chloride channel-associated cancers and present recent advances in understanding the environments of various types of cancer caused by the participation of many chloride channel or transporters families and discuss the challenges and potential targets for cancer treatment. The modulation of chloride channels/transporters might promote new aspect of cancer treatment strategies.

## 1. Chloride Transport

Electrolytes, such as charged anion chloride, drive cellular electrical shifting energy, which mediates various cellular processes. Intracellular chloride ions are abundant (5–40 mM) as are sodium ions [1]. The movement of chloride is considered to regulate cellular membrane potential, cellular volume, and electrostatic compensation as well as maintain the pH of cellular or intra-organelles such as lysosomes. In addition to its classical roles, chloride channels participate in modulation of the cellular fate and motility of cancer cells. Therefore, the finding that these channels function in malignant conditions beyond simply transporting chloride is meaningful. This review summarizes the prevalence and roles of several families of chloride channels/transporters associated with malignant environments and might facilitate a better understanding of cancer and aid in the identification of potential targeted anticancer agents with scope of chloride channel/transporter-based tumorigenesis.

## 2. Membrane-Associated Components of Chloride Movement

### 2.1. Transmembrane Member 16A

Transmembrane member (TMEM) 16A (calcium-activated chloride channel; anoctamin 1, ANO1) transports chloride and bicarbonate and plays a role in the proliferation and development of malignant cell types. The expression of TMEM16A has been identified in a broad range of cancers such as non-small cell lung cancer (NSCLC) [2], pancreatic cancer [3], prostate cancer [4], breast cancer [5], colorectal carcinoma [6], gastric cancer [7], glioma [8], glioblastoma [9], esophageal cancer [10], lung cancer [11], hepatocellular carcinoma (HCC) [12], liposarcoma [13], leiomyosarcoma [14], salivary gland cancer [15], and chondroblastoma [16]. Cellular specific mechanism of TMEM16A is extensively reviewed and highlighted in various cancers [17,18]. Briefly, TMEM16A positively correlates with epidermal growth factor receptor (EGFR) expression in tumor development [2], and both TMEM16A and EGFR are found in NSCLC tissues. Tumor, node, metastasis (TNM) stage 3 + 4 primary NSCLC is positive for TMEM16A and EGFR [2]. Thus, TMEM16A is considered as a potential diagnostic marker for lung cancer. Treatment of TMEM16A inhibitor T16Ainh-A01 or knockdown of TMEM16A inhibits the cellular proliferation and invasion by attenuating EGFR phosphorylation in H1299 lung cancer cells [19]. Knocking down TMEM16A attenuates proliferation and migration by inhibiting phosphoinositide 3-kinase/protein kinase B (PI3K/PKB) and mitogen-activated protein kinase (MAPK) pathways in HCC, HepG2, and SMMC7721 cells [20]. Colorectal cancer (CRC) and HCT116 and DLD-1 cells also express abundant TMEM16A, which is a prognostic factor for patients with CRC [21]. MicroRNA-132 (miR-132) has been identified in nerve tissues of mice, humans, zebrafish, and cattle [22] and it functions as a tumor suppressor in various cancers to prevent metastasis and proliferation [23]. Attenuating TMEM16A through miR-132 decreases cellular proliferation, invasion, and liver metastasis [21]. In addition, bestrophin-1 is also considered as a putative calcium-activated chloride channel such as TMEM16A in epithelial cells such as the cystic fibrosis pancreatic duct cell line, CFPAC-1 [24,25,26,27,28] and it enhances calcium signaling and volume regulation in CRC T-84 cells by participating in proliferation [24]. Bestrophin-1 is also associated with the proliferation of oral squamous cell carcinoma (OSCC) HST-1 cells [27]. Although bestrophin-1 consists of calcium-activated chloride channels that are dependent on or independent of other proteins, it interacts with TMEM16A in normal tissues [29,30,31].

### 2.2. TMEM206

An acidic milieu is involved in various diseases, such as ischemia, cancer development, and inflammation [32]. Acid-sensitive chloride channels (also known as proton-activated chloride channels, PAC; TMEM206) are expressed in normal and malignant tissues. Protein profiling has revealed that colorectal, breast, and hepatic cancer cells have increased amounts of TMEM206, which plays a key role in cellular responses to acidic conditions [32,33]. Consistent with this concept, silencing TMEM206 attenuates acid-mediated cell death and alleviates acidosis-associated pathologies such as ischemic stroke. The cytoplasmic expression of TMEM206 is associated with CRC development and proliferation. The CRC cell lines SW480 and HCT-116 overexpress TMEM206, which results in enhanced cellular migration, invasion, and proliferation via AKT/ERK phosphorylation [32]. Although the precise mechanism of action of TMEM206 in other cancerous tissues remains unknown, TMEM206 could be considered as a diagnostic marker for CRC.

### 2.3. Calcium-Activated Chloride Channel Regulators

Calcium-activated chloride channel regulators (CLCAs) modulate chloride in epithelia, play critical roles in transporting electrolytes including chloride, modulate function of TMEM16A and its adhesion molecules, and negatively regulate cancer development. Nasopharyngeal, breast, and colorectal cancers have low levels of CLCAs [34,35,36,37,38]. The CLCA1 protein is primarily expressed in the small intestine, colon, and appendix. The expressions of CLCA1 and CLCA4 are decreased in intestinal tissues from patients with CRC, the CRC cell lines SW620 and LOVO, and in hormone receptor-positive breast cancer cell line MCF7 cells [35,37,38]. CLCA1 is negatively involved in the differentiation of intestinal Caco-2 cells [38]. Low levels of CLCA2 mRNA and protein have been identified in nasopharyngeal carcinoma (NPC) S18 and 5-8F cells, whereas overexpressed CLCA2 inhibits FAK/ERK signaling in these cells [34]. Transduction with p53 induces increased CLCA2 and inhibits the proliferation of breast cancer MCF10A and BT549 cells [36]. Overexpressed CLCA4 inhibits the epithelial-mesenchymal transition (EMT), which is involved in the migration and invasion of CRC cells, whereas CLCA4 depleted by shRNA enhances cellular migratory and invasive ability through enhanced EMT in human mammary epithelial cells [35,37]. Although further evidence is needed, CLCA levels could be considered as potential diagnostic biomarkers.

### 2.4. Chloride Intracellular Channels

Chloride intracellular channel 1 (CLIC1; also known as NCC27) belongs to the highly conserved CLIC family of chloride ion channels [39]. It can reside in the cytoplasm and temporarily in plasma and internal cell membranes [40]. CLIC1 participates in various cellular functions, including the maintenance of pH homeostasis, cell survival, cell cycle regulation, cell volume regulation, membrane potential modulation, and organelle acidification [40,41,42,43,44,45,46,47,48]. This channel is upregulated in various cancer type such as prostate [46], gallbladder (GBC) [48], colon cancer [47], gastric [49], clear cell renal cell carcinoma [50], and glioblastoma stem cells [51,52]. Overexpressed CLIC1 in patients with HCC [53] positively correlates with HCC proliferation and metastasis [54]. CLIC1 participates in hypoxia-induced colonic carcinoma metastasis via the MAPK/ERK pathway [45]. Moreover, CLIC1 is recruited to the plasma membrane in response to chemotaxis, such as directional treatment with epidermal growth factor (EGF) and mechanotaxis and its ectopic expression of CLIC1 enhances migratory apparatus such as lamellipodia and invadopodia [54]. Hypoxia-induced tumor cells possess irregular microvascular networks and blood flow [55] and can be transformed to promote cancer metastasis [56]. Mechanistically, limited blood perfusion or altered flow due to hypoxic conditions might contribute to the migration and invasion of cancer cells [57]. Upregulated CLIC1 expression correlates with lymph node metastasis and lymphatic invasion [49] as well as lung cancer migration and invasion [40]. Cell growth is promoted by CLIC1 via the MAPK/ERK pathway in prostate cancer [46] and CLIC1 is expressed in pancreatic ductal adenocarcinoma (PDAC) [58] where it plays an important role in promoting cancer cell survival, proliferation, and invasion [46,59]. In various regulatory processes involving CLIC1, small interfering (si)RNAs of CLIC1 induce the downregulation of cell proliferation, growth, and invasiveness of pancreatic cancer cell lines such as PANC-1 and MIAPaca-2 compared with control cells [58,60]. Furthermore, CLIC1 is associated with proteasome activator 28 β (PA28 β), and its specific siRNA downregulates CLIC1 in gastric cancer [61]. A regulatory volume decrease (RVD) is a critical process in cancer cell motility, such as migration and invasion [62]. The CLIC1 inhibitors IAA94 or CLIC1-specific siRNAs ameliorate the RVD and decrease the migration and invasion of CRC LOVO and HT-29 cell lines [45]. The expression of CLIC1 at the mRNA and protein levels is downregulated by miR-124 transfection in the hepatic cancer cell line HepG2 and by hsa-miR-372 transfection in the GBC cell lines G-415, OCUG-1, and SGC-996 [63,64]. The downregulation of CLIC1 reduces cell migration and invasion [63]. The siRNA of CLIC1 enhances expression of the tumor metastasis-related genes annexin A7 and gelsolin, the knockdown of which increases CLIC1 expression in mouse HCC Hca-F and Hca-P cell lines, suggesting that CLIC1 interacts with annexin A7 and gelsolin, and mediates tumor cell migration, invasion, and metastasis [65]. In addition, biguanide-related drugs such as metformin, morocydine, and proguanil inhibit CLIC1 current and dysregulate proliferation and invasiveness in glioblastoma stem cells [66]. Cells and tumors expressing Rab25 have abundant CLIC3 that co-localizes with active integrin α5β1 in ovarian cancer A2780 cells [67]. High CLIC4 expression in PDAC together with Indian hedgehog is a proposed metastatic marker of PDAC [68]. In contrast, CLIC2 is expressed in non-cancerous masses and is a potent regulator of tight junctions. The expression of CLIC2 and tight junction proteins is upregulated in non-cancer, compared with cancer cells, and CLIC2 regulates the formation of tight junction proteins such as claudin 1, claudin 5, zonula occludens-1, and occludin [69]. CLIC1 and other types of chloride channels could be potential treatment strategies for cancer and should be considered as novel diagnostic and therapeutic targets for prostate, gastric, gallbladder, colon, pancreas, lymphatic, and lung cancers.

### 2.5. Voltage-Gated Chloride Channels

Voltage-gated chloride channel 3 (CLC-3; also known as CLCN3) is expressed in cell membranes and intracellular vesicles where it exchanges chloride for hydrogen. CLC-3 protein is expressed in prostate carcinoma [70], nasopharyngeal [62], neuroendocrine [71], and brain cells [72] and is significantly overexpressed in HCC, compared with normal control tissues [73]. Moreover, upregulated CLC-3 is associated with HCC tumor size and prognosis [73]. Overexpressed CLC-3 protein participates in cell proliferation and migration. The regulation of cell volume by CLC-3 is involved in the development and metastasis of NPC and prostate cancer [62,70,74]. Signaling by Wnt/β-catenin contributes to metastasis and adhesion by regulating the EMT process in tumorigenesis [75,76]. The expression of CLC-3 is more abundant in tissues at the late stage of CRC and in the CRC LOVO and SW620 compared with that in normal cells. SiRNA-CLC-3 (siCLC-3) inhibits CRC cell viability, proliferation, and metastasis by inhibiting Wnt/β-catenin signaling, whereas the Wnt/β-catenin activator lithium chloride rescues the effect of siCLC-3 [77]. CLC-3 could be a prognostic marker for HCC, CRC, NPC, and prostate cancer. Patients with breast cancer are treated with tamoxifen, a non-steroidal anticancer agent [78] that inhibits the migration, chloride current, and volume regulatory mechanisms in HCC MHCC97H cells in vitro [79]. The activator of protein kinase C (PKC) phorbol-12-myristate-13 acetate (PMA) inhibits PKC expression in the presence of tamoxifen and reduces the migration of cells with CLC-3 knockdown, suggesting that CLC-3 is involved in the mechanism of anticancer drug and cellular volume regulation [80]. CLC-4 is expressed on the cell surface and intracellular endosomal membranes in the CRC cell lines RKO and LS174 [81]. The migration and invasion of CRC cells is reduced by CLC-4 siRNA or shRNA [82]. Incomplete glucose metabolism results in increased intracellular proton concentrations driven by CLC-4, which maintains a neutral intracellular pH and the essential proton extrusion mechanism [83]. The regulation of pH by CLC-4 in the endosomal compartment might participate in promoting invasive ability. Active acidification of large intracellular endosomal vesicles by a vacuolar H^+^-ATPase promotes proteolysis of the extracellular matrix, activates pro-cathepsin D which is activated when acidic condition is triggered, and facilitates proteolytic function on the basement membrane [82,84]. pH regulation in the cytosol and intracellular organelles in RKO cells overexpressing CLC-4 results in resistance to acid-induced cytotoxicity, which is similar to an acidic tumor microenvironment and the enhanced ability of colon cancer cells to migrate [82].

### 2.6. Cystic Fibrosis Transmembrane Conductance Regulator

Cystic fibrosis transmembrane conductance regulator (CFTR) is a cAMP-activated chloride channel that regulates the balance of electrolytes in the respiratory and endocrine systems, exocrine glands, and other tissues. Malfunctioning and/or abnormal expression of CFTR have been found in various types of cancer. The upregulated expression of CFTR is associated with an invasive phenotype in cervical and ovarian carcinomas [85,86]. Conversely, the mRNA and protein expression of CFTR are reduced in NPC 5-8F, 6-10B, and HNE-1, compared with normal cells, whereas CFTR knockdown increases NPC cell migration and invasion [87]. Enhanced CFTR protein expression in NPC 5-8F cells increases epithelial markers such as occludin and E-cadherin, and attenuates the mesenchymal marker smooth muscle actin [87]. In addition, protein expression of CFTR is decreased in CRC, compared with normal tissues [88]. However, CFTR mRNA overexpression decreases cell proliferation, migration, and invasion in the CRC cell lines HCT116 and CaCo-2 [88]. The results of studies on the roles of CFTR in cancer have been contradictory. Enhanced CFTR expression inhibits various cancerous processes such as EMT in breast carcinoma [89], lung cancer [90], NPC [87], endometrial carcinoma cells [91], prostate cancer [92], and intestinal carcinoma [93]. Cisplatin increases CFTR expression and enhances chemoresistance and the cell viability of prostate cancer tissues compared with chemo-sensitive prostate cancer tissues in vivo and LNCaP cells in vitro [94]. Nicotine is a potential cause of lung cancer and a progressive enhancer of adenocarcinoma cells that inhibits the CFTR protein expression in A549 cells [95]. Although the CFTR gene could act as a tumor suppressor, its roles in various cell types and cancer cells need to be defined.

### 2.7. Voltage-Dependent Anion Channels

The expression of voltage-dependent anion channels (VDACs) on the mitochondrial membrane of all eukaryotes, including mammals [96], is increased in various tumor tissues, such as carcinoma of the breast [97], colon [98], thyroid gland [99], lung [100], pancreas [101], and liver [102] compared with that in normal tissues. The VDAC1, 2, and 3 isotypes of these channels play different roles; VDAC1 and VDAC2 participate in pore formation within the mitochondrial membrane [96] and VDAC3 participates in the regulation of mitochondrial membrane potential [103]. The expression of VDAC is associated with neurodegenerative disorders and muscular and myocardial diseases including various types of cancers [104]. The progression of tumorigenesis is decreased in HeLa cells with depleted VDAC1 [105]. The expression of VDAC1 is more abundant in cancer, A549, and HeLa cells, than in normal WI-38 fibroblasts derived from lung tissue [106], HCC tissues, HepG2 and SMMC7721 cells, as well as lung adenocarcinoma tumors [107]. Small interfering RNA-VDAC1 and miR-7 downregulate cell growth, proliferation, migration, and invasion in HCC tissues [102], lung cancer A549 cells [100], and cervical cancer HeLa cells [105]. Furthermore, miR-490-3p is significantly associated with the carcinogenesis of various cancers [98], and it can regulate the growth and EMT of HCC cells [108] and the invasiveness of triple-negative breast cancer cells, MDA-MB-231, and MDA-MB-436 [109]. MiR-490-3p downregulates VDAC1 through the mammalian target of rapamycin (mTOR) pathway in CRC tissues and cell lines [98]. The expression of VDAC2 is upregulated in melanoma cells and HCC cell lines such as HepG2 [110] but downregulated in glioma stem cells [111] and it plays an anti-apoptotic role in primary cultured mouse embryonic fibroblasts. Although its different role of VDACs is defined, precise roles of VDAC family in different cancers remain unresolved and await identification in future studies.

### 2.8. Volume-Regulated Anion Channel

Volume regulation is critical function to maintain cellular fate. Volume-regulated anion channels (VRACs; also called volume-sensitive organic osmolyte anion channel or swelling-induced chloride current ICl_swell_) are considered as regulatory channels of cellular volume [112]. VRAC is involved in the RVD and regulates proliferation of nasopharyngeal carcinoma cell [113,114,115], OSCC HST-1 cells [27], and gastric cancer [116]. Inhibited VRAC by 4-(2-Butyl-6,7-dichlor-2-cyclopentyl-indan-1-on-5-yl) oxybutyric acid (DCPIB) reduces proliferation, migration, and invasion of glioblastoma U251 and U87 cells [117]. Leucine-rich-repeat-containing 8A (LRRC8A, also called SWELL1) is component protein of VRAC. LRRC8A expresses in HCC tissues and induced cellular proliferation and migration in HCC SMMC-7721, Sk-hep-1, Huh7, and HCCLM3 cells [118]. Moreover, survival of cisplatin-resistant A549 cells or A2780 cells is modulated by the LRRC8A [119,120]. Although dual function of VRAC components LRRC8A and LRRC8D on drug resistance has been addressed [121], VRACs may be associated with chemo-resistant mechanism and are needed to verify its precise mechanism in various cancers.

### 2.9. Chloride-Bicarbonate Exchangers

Chloride-bicarbonate (CB) exchangers consist of solute carrier (SLC) families, including anion exchangers (AEs) and SLC26As. The CB exchangers mediate the electroneutral or electrogenic exchange of bicarbonate for chloride (respective stoichiometry of chloride: bicarbonate, 1:1 or 1:2) and are associated with the regulation of intracellular pH. The anion exchangers AE1, AE2, AE3, and AE4 [122,123,124,125] are expressed in various tissues and localize in the plasma membrane. Both AE1 and AE2 are involved in cancer whereas other AEs are unknown. Histologic findings have shown that AE1 is expressed in the cytoplasm of gastric cancer cells [126,127]. Knockdown of AE1 induces the release of p16INK4A and inhibits gastric cancer growth [126,128]. The expression of AE1 positively correlates with cancer size and metastasis [127]. Although the modulation of AE1 expression is poorly verified, miR-24-mediates AE1 attenuation in gastric cancer cells [129]. The expression of AE1 is associated with tumor progression through crosstalk with MAPK and hedgehog signaling pathways in esophageal carcinoma [130]. Although AE2 is expressed in most tissues, it has been addressed that the AE2 gene is highly expressed in HCC cells, gastric, and colorectal cancers [131,132,133,134,135]. An antisense oligonucleotide of AE2 inhibits HCC progression [135]. AE2 is also expressed in ovarian cancer and it participates in tumorigenesis through activation of the mTOR/p70S6K1 pathway [136]. AE2a that is localized in the Golgi apparatus is involved in the malignancy of SW-48 CRC cells [137]. We previously found that the anti-alcoholism agent disulfiram exerts anticancer effects and attenuates the membrane expression of AE2 and the supportive enzyme carbonic anhydrase XII through disturbed homeostatic pH regulation in lung cancer cells [138]. Our Western blotting findings in vitro also showed that disulfiram attenuated AE2 protein expression and CB exchanging activity in the breast cancer cell lines MDA-MB-231 and MCF-7 [138]. Although anti-cancer effect of disulfiram is addressed, the clinical relevance of AE2 in these cancers and the clinical effects of disulfiram should be verified more precisely in other types of cancer. The role of the SLC26 family in cancerous tissues or mechanisms is not well understood. Chondrodysplasias, chronic chloride diarrhea, and deafness in humans are linked to SLC26A2, SLC26A3, and SLC26A4, respectively [139]. SLC26A3 participates in chloride homeostasis and interactions with CFTR [140]. The expression of SLC26A3 is downregulated and modulated by the stomach-specific 18 kDa antrum mucosal protein in gastric cancer cells [141]. Analyses of gene sets and protein-protein networks have revealed a relationship between SLC26A6 and HCC [142]. However, further detailed investigation is needed to reveal whether any members of the SLC26A family have potential to serve as diagnostic or prognostic markers in cancerous tissues.

## 3. Perspectives

The modulation of chloride channels/transporters in cancers has been considered as challenging issues. Complicated ion movements, crosstalk among channels/transporters, difficulties with experimental implementation, and/or compensation of unidentified mechanisms hamper investigations into ion channels/transporters. For example, studies of intracellular ion channels, not plasma membrane-associated channels, have struggled with technical limitations in terms of verifying physiological roles [143].

Recently, advanced therapeutic candidates of chloride transport were reviewed in various organ diseases such as inflammatory lung disease, osteoporosis, dry eye disorders, hypertension, polycystic kidney disease, and kidney stone [112]. Although approved drugs or preclinical trials of chloride transporters are suggested [112], approaches in multiple cancers remain mostly unknown. Thus, this review would be beneficial to expand developed application of trials through an overview of chloride channel/transporter-associated cancers. Interestingly, several chloride channels/transporters are involved in at least two types of cancer (Figure 1 and Table 1). Although shared chloride channels might not reflect similarity among types of cancer, multiple facets should be considered to verify the roles of transporters and convergent regulation in normal and cancer tissues. Moreover, identifying crosstalk among cancers through shared chloride channels might provide valuable clues to metastatic cancer. Although accumulating information about chloride channels and transporters provides both challenges to investigator and potential therapeutic targets against cancer, mechanism of channels and transporters requires further verification. In addition, application of developed chloride transport modulators for cancers should be beneficial to verify off-target effect and toxicity on normal tissue for cancer treatment.

## Author Contributions

H.J.K., P.C.-W.L. and J.H.H. contributed to the conceptualization and design of the review. H.J.K. collected the information, drafted the article, prepared the figure, and critically revised the manuscript for important intellectual content. J.H.H. and P.C.-W.L. All authors have read and agreed to the published version of the manuscript.

## Figures and Tables

**Figure 1 cancers-14-00856-f001:**
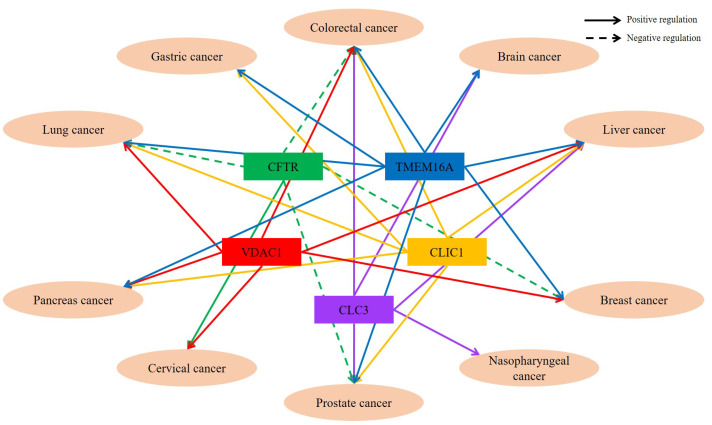
Schema of chloride channel-associated cancers. Chloride channels are involved in at least two types of cancers. Identification of shared chloride channels might provide clues to identify metastatic cancer. VDAC1: Voltage-dependent anion channel 1, TMEM: Transmembrane member, CLC: Voltage-gated chloride channels, CLIC: Chloride intracellular channels, CFTR: Cystic fibrosis transmembrane conductance regulator.

**Table 1 cancers-14-00856-t001:** Diseases-related chloride channels/transporters and their functions.

Diseases	Chloride Channels/Transporters	Function	Reference
Lung cancer	TMEM16A	Cell proliferation and development to malignant tumor Positive correlation between TMEM16A and EGFR	[2,11,19]
	TMEM206	Enhanced cellular migration, invasion, and proliferation	[32]
	CLIC1	Enhanced cellular migration and invasion	[40]
	CFTR	Inhibited various cancer-related processes, such as EMT	[90,95]
	VDAC1	Cell growth, proliferation, migration, and invasion	[100,107]
Pancreatic cancer	TMEM16A	Cell proliferation and development to malignant tumor	[3]
	CLIC1	Cell proliferation, growth, and invasiveness	[58,60]
	CLIC4	Proposed metastatic marker	[68]
	VDAC1	Cell growth, invasion, and migration	[101]
Hepatic cancer	TMEM16A	Cell proliferation and development to malignant tumor	[12,20]
	CLIC1	Promoted cancer cell survival, proliferation, and invasion	[53,54,59,63,65]
	CLIC2	Correlated with the tight junction	[69]
	CLC3	Related to tumor size and prognosis marker Cell volume regulation	[73,79,80]
	VDAC1	Regulated cell growth and EMT Cell growth, proliferation, migration, and invasion	[102,108]
	VDAC3	Regulation of mitochondrial membrane potential	[103]
	AE2	Inhibited proliferation and viability of cancer cells	[131,135]
	SLC26A6	Diagnostic or prognostic biomarker for cancer	[142]
Nasopharyngeal cancer	CLCA2	Negatively regulate cancer development Regulate FAK/ERK signaling	[34]
	CLC3	Correlation between cell volume regulation and cancer development and metastasis	[62,74]
	CFTR	Enhanced cell migration and invasion Regulate EMT	[87]
	VRAC	Regulates proliferation	[115]
Breast cancer	TMEM16A	Cell proliferation and development to malignant tumor	[5]
	CLCA2	Negatively regulate cancer development	[36]
	CLCA4	Negatively regulate cancer development Decreased EMT	[37]
	CFTR	Inhibited various cancer-related processes, such as EMT	[89]
	VDAC1	Promoted cell proliferation	[97]
Colorectal cancer	TMEM16A	Cell proliferation and development to malignant tumor	[6,21]
	Bestrophin-1	Involved in cellular proliferation	[24]
	CLCA1	Negatively regulate cancer development	[38]
	CLCA4	Negatively regulate cancer development Decreased EMT	[35]
	CLIC1	Enhanced cellular migration and invasion	[45,47]
	CLC3	Regulate cell viability, proliferation, and metastasis	[77]
	CLC4	Regulate invasion, migration, and pH of cytosolic and intracellular organelles	[81,82]
	CFTR	Decreased cell proliferation, migration, and invasion	[88]
	VDAC1	Regulate apoptosis, proliferation, migration, and invasion	[98]
	AE2	Contributed to progression of cancer Promote tumor cell malignancy Responsible for elevated Golgi resting pH	[132,137]
Prostate cancer	TMEM16A	Cell proliferation and development to malignant tumor	[4]
	CLIC1	Promoted cancer cell survival, proliferation, and invasion	[46]
	CLC3	Correlation between cell volume regulation and cancer development/ metastasis	[70]
	CFTR	Inhibited various cancer-related processes, such as EMT Enhanced chemo-resistance	[92,94]
Gallbladder cancer	CLIC1	Enhanced cellular migration and invasion	[48,64]
Gastric cancer	TMEM16A	Cell proliferation and development to malignant tumor	[7]
	CLIC1	Enhanced cellular migration and invasion	[44,49,61]
	AE1	Involved in release of p16INK4A and cell growth Positive correlation with cancer size and metastasis	[126,127,128,129]
	AE2	Contributed to progression of cancer	[133,134]
	SLC26A3	Interact with AMP18	[141]
	VRAC	Regulates proliferation	[116]
Brain tumor	TMEM16A	Cell proliferation and development to malignant tumor	[8,9]
	CLC3	Enhanced cell invasion	[72]
	VDAC2	Regulator for the metabolic reprogramming	[111]
Clear cell renal cell carcinoma	CLIC1	Cell invasion	[50]
Neuroendocrine tumor	CLC3	Enhanced resistance to anticancer drug	[71]
Esophageal cancer	TMEM16A	Cell proliferation and development to malignant tumor	[10]
	AE1	Related with tumor progression through the crosstalk with MAPK and hedgehog signaling pathways	[130]
Liposarcoma	TMEM16A	Cell proliferation and development to malignant tumor	[13]
Leiomyosarcoma	TMEM16A	Cell proliferation and development to malignant tumor	[14]
Salivary gland cancer	TMEM16A	Cell proliferation and development to malignant tumor	[15]
Chondroblastoma	TMEM16A	Cell proliferation and development to malignant tumor	[16]
Hydatidiform moles	CLIC1	Enhanced cellular migration and invasion	[43]
Ovarian cancer cells	CLIC3	Co-localized with active integrin α5β1 Cell migration and invasion	[67]
	CFTR	Enhanced cell invasion and migration	[85]
	AE2	Involved in tumorigenesis through activation of the mTOR/p70S6K1 pathway	[136]
Cervical cancer	CFTR	Enhanced cell invasion and migration	[86]
	VDAC1	Regulate progression of tumorigenesis	[105]
Endometrial cancer	CFTR	Inhibited various cancer-related processes, such as EMT	[91]
	VDAC1	Pore-forming role in mitochondrial membrane Cell growth, proliferation, migration, and invasion	[96]
Oral squamous cell carcinoma	Bestrophin-1	Involved in proliferation	[27]
	VRAC	Regulates proliferation	

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
