# Peer review of "Chloride Channels and Transporters: Roles beyond Classical Cellular Homeostatic pH or Ion Balance in Cancers"

_cancers, 2022, doi:10.3390/cancers14040856_

Round 1

Reviewer 1 Report

The review, titled “Chloride channels: Roles beyond classical cellular homeostatic pH or ion balance in cancers”, summarizes the role of several families of chloride channels associated with cancer environment, in addition the modulation of chloride channels/transporters might promote new aspect of cancer treatment strategies.  The scientific collect is very interesting, however, some problems, as indicated below, should be addressed before the document can be considered for publication in this journal. This version of the manuscript is not enough complete.

Here, I present all my objections in details.

Major revision:

English language and style are fine, minor spell check is required to ensure that an international audience can clearly understand your text. In general, I suggest to review the style of the manuscript according to the guidelines of the journal.

In the title, the authors indicated "chloride channels", but in the full text they also evidence the importance of chloride-associated transporters. Thus, I suggest to modify the title, also because there is a notable difference between channels and ion transporters.

I suggest to modify the section 2.2 with 2.3 (reverse paragraphs).

The authors should add recent references, no evidence of 2021 are present in the text.

Author Response

Dear reviewer and editor,

Before addressing each of the comments below, we appreciate the reviewers for the valuable comments and careful consideration. We obviously have needed to quote all sources correctly and done so at the places where we had missed before. In addition, the manuscript has been edited to make appropriate information to this body of work.

Responses to comments of reviewer as below:

Reviewer1

The review, titled “Chloride channels: Roles beyond classical cellular homeostatic pH or ion balance in cancers”, summarizes the role of several families of chloride channels associated with cancer environment, in addition the modulation of chloride channels/transporters might promote new aspect of cancer treatment strategies.  The scientific collect is very interesting, however, some problems, as indicated below, should be addressed before the document can be considered for publication in this journal. This version of the manuscript is not enough complete.

Here, I present all my objections in details.

Major revision:

English language and style are fine, minor spell check is required to ensure that an international audience can clearly understand your text. In general, I suggest to review the style of the manuscript according to the guidelines of the journal.

In the title, the authors indicated "chloride channels", but in the full text they also evidence the importance of chloride-associated transporters. Thus, I suggest to modify the title, also because there is a notable difference between channels and ion transporters.

-Response: We appreciate your valuable comment and we agreed your comment. We modified the title, included channels and transporters, to “Chloride channels and transporters: Roles beyond classical cellular homeostatic pH or ion balance in cancers”

I suggest to modify the section 2.2 with 2.3 (reverse paragraphs).

-Response: We appreciate your valuable comment and we rearranged the section 2.2 and 2.3.

The authors should add recent references, no evidence of 2021 are present in the text.

-Response: We appreciate your valuable comment and we added recent articles in the text.

Reviewer 2 Report

The review “Chloride channels: Roles beyond classical cellular homeostatic pH or ion balance in cancers” by Kim et al, provides a timely update on the relatively less appreciated functions of chloride channels. The authors have done an appreciable coverage of the topic and I have nothing to take away from this review. I believe this review will serve as an important point of reference for future studies. On that regard, I request the authors to attend to the following, which is essential to further improve the quality of the review:

  1. What did the authors mean by “difficulties with experimental implementation were disturbed” in line 3 of simple summary? Is that a typo?
  2. Line 129 should be rewritten as it appears to indicate PANC1 and MIA-PACA cells as siRNAs: “…CLIC1 small interfering (si)RNAs such as PANC-1 and MIAPaca-2 induce..”
  3. Roles of CLIC channels in cancer stem cells should be incorporated in the corresponding sub-section.
  4. The authors should incorporate Cl channels of the type - volume regulated anion channels (VRACs) and their role in chemoresistance in a sub-section.

Author Response

Dear reviewer and editor,

Before addressing each of the comments below, we appreciate the reviewers for the valuable comments and careful consideration. We obviously have needed to quote all sources correctly and done so at the places where we had missed before. In addition, the manuscript has been edited to make appropriate information to this body of work.

Responses to comments of reviewer as below:

Reviewer2

The review “Chloride channels: Roles beyond classical cellular homeostatic pH or ion balance in cancers” by Kim et al, provides a timely update on the relatively less appreciated functions of chloride channels. The authors have done an appreciable coverage of the topic and I have nothing to take away from this review. I believe this review will serve as an important point of reference for future studies. On that regard, I request the authors to attend to the following, which is essential to further improve the quality of the review:

  1. What did the authors mean by “difficulties with experimental implementation were disturbed” in line 3 of simple summary? Is that a typo?

-Response: We appreciate your valuable comment and we edited the sentence in simple summary.

  1. Line 129 should be rewritten as it appears to indicate PANC1 and MIA-PACA cells as siRNAs: “…CLIC1 small interfering (si)RNAs such as PANC-1 and MIAPaca-2 induce..”

-Response: We appreciate your valuable comment and we rewrote the sentence to “CLIC1 small interfering (si)RNAs induce the downregulation of cell proliferation, growth, and invasiveness of pancreatic cancer cell lines such as PANC-1 and MIAPaca-2 com-pared with control cells”

  1. Roles of CLIC channels in cancer stem cells should be incorporated in the corresponding sub-section.

-Response: We appreciate your valuable comment and included CLIC channels in cancer stem cells (Section 2.4)

  1. The authors should incorporate Cl channels of the type - volume regulated anion channels (VRACs) and their role in chemoresistance in a sub-section.

-Response: We appreciate your valuable comment and added VRAC section (Section 2.8).

Reviewer 3 Report

Review article ID cancers-1558439, summarizes recent findings in the involvement of chloride channels in different types of cancer. The Manuscript is well written and clearly organized, but from my viewpoint it should consider also the recent advancements in the development of small molecules targeting chloride channels. This would surely broaden audience and add useful information. Despites the  modulation of these protein appears complicated, as Authors state in the Perspectives paragraph, significant advancements have been made in this field and a small discussion about the most important and recent molecules should be added. (see for example Am J Physiol Cell Physiol 321: C932–C946, 2021. doi:10.1152/ajpcell.00334.2021)

Author Response

Dear reviewer and editor,

Before addressing each of the comments below, we appreciate the reviewers for the valuable comments and careful consideration. We obviously have needed to quote all sources correctly and done so at the places where we had missed before. In addition, the manuscript has been edited to make appropriate information to this body of work.

Responses to comments of reviewer as below:

Reviewer3

Review article ID cancers-1558439, summarizes recent findings in the involvement of chloride channels in different types of cancer. The Manuscript is well written and clearly organized, but from my viewpoint it should consider also the recent advancements in the development of small molecules targeting chloride channels. This would surely broaden audience and add useful information. Despites the  modulation of these protein appears complicated, as Authors state in the Perspectives paragraph, significant advancements have been made in this field and a small discussion about the most important and recent molecules should be added. (see for example Am J Physiol Cell Physiol 321: C932–C946, 2021. doi:10.1152/ajpcell.00334.2021)

-Response: We appreciate your valuable comment and information. Your suggestion is informative. This review would be beneficial to expand developed application of trials through an overview of chloride channel/transporter-associated cancers. We added small discussion about the recent molecules as you recommended and added recent articles in the text.

Round 2

Reviewer 1 Report

The authors have satisfyingly addressed all concerns and suggestions. However, I have still a suggestion. I suggest to add the following reference in the section "Voltage-dependent anion channels" (DOI: 10.3390/ijms22168359).

Reviewer 3 Report

Authors have partially addressed my issue, and the overall manuscript was quite ameliorated. In my opinion it can be now published in its present form